behaviour/ecology/evolution

aggregation, antlion, competition, *Euroleon nostras*, thermal biology, trap building

**Author for correspondence:**
Krzysztof Miler
e-mail: miler@isez.pan.krakow.pl

# Past thermal conditions affect hunting behaviour in larval antlions

Krzysztof Miler[1] and Marcin Czarnoleski[2]

[1]Institute of Systematics and Evolution of Animals, Polish Academy of Sciences, Sławkowska 17, 31-016, Kraków, Poland
[2]Institute of Environmental Sciences, Faculty of Biology, Jagiellonian University, Gronostajowa 7, 30-387 Kraków, Poland

 KM, 0000-0001-7684-0629; MC, 0000-0003-2645-0360

Some sit-and-wait predators, such as antlion larvae, construct traps to capture passing prey. The location of these traps depends on many abiotic and biotic factors, including temperature and the presence of conspecifics, which probably stimulate behaviours that minimize the costs and maximize the benefits of trap building. Here, we exposed second instar antlion larvae to elevated temperatures of 25°C (mild treatment) or 31°C (harsh treatment) for one month and then transferred them to common conditions (20°C) to examine the effects of previous thermal treatment on aggregation tendency and trap size. We predicted that antlions that experienced harsh conditions would subsequently increase the neighbouring distance and trap diameter to reduce competition with conspecifics and improve prey capture success, compensating for past conditions. In contrast with these predictions, antlions exposed to harsh conditions displayed a trend in the opposite direction, towards the decreased neighbouring distance. Furthermore, some of these antlions also built smaller traps. We discuss possible reasons for our results. The effects of previous thermal exposure have rarely been considered in terms of trap construction in antlions. Described effects may possibly apply to other sit-and-wait predators and are significant considering that many of these predators are long-lived.

## 1. Background

Predatory species have evolved a broad spectrum of hunting strategies, from highly mobile predators that chase escaping prey at record-breaking speeds to sedentary predators that remain in place to ambush passing prey. Some sit-and-wait predators extend their phenotypes by constructing hunting structures, such as webs by spiders or pitfall traps by antlion larvae [1]. The predation success of trap-building predators

depends greatly on the trap location, and not surprisingly, the choice of a site for trap establishment depends on a variety of environmental properties, including disturbance level and prey availability for spiders [2,3] and substrate type and conspecific density for antlions [4,5]. Microhabitat properties that affect trap building in antlions have been a subject of scientific investigation for many years [6]. However, it is still difficult to generalize the significance of many potential drivers of trap construction in this group of insects, probably due to the species-specific characteristics of these effects. Indeed, some antlion species exclusively inhabit shaded areas, whereas others inhabit well-lit areas; the two groups differ in thermal tolerance [7,8] and the type and abundance of prey [9,10] and show different responses to some fundamental environmental factors, such as thermal conditions and feeding rates [11]. For example, shade-occupying *Myrmeleon hyalinus* antlions are more susceptible to starvation than *Cueta lineosa* antlions, which inhabit open areas exposed to direct sunlight; this explains why the former species prefers sites with more abundant prey [12].

Preferable conditions, whatever they might be, are usually limited, resulting in competition among conspecifics, which becomes increasingly intense with an increasing density of competitors. Antlions exposed to high conspecific densities reduce trap construction activity [13,14], establish smaller traps [15–17] and increase the frequency of trap relocation [18–20]. Thus, conspecific density seems to be an important factor affecting hunting and trap construction in antlions. Nevertheless, the response to a high density of conspecifics might be mediated by other factors. For example, high density does not increase the frequency of relocation when prey is abundant [21] despite increased living costs associated with aggregation, such as the need to frequently maintain traps exposed to constant disturbances by neighbours and/or reduced size of traps [17,22–27]. There is evidence to suggest that a larger distance between traps leads to lower interference between antlions and lower relocation propensity (see [18]) and that larger traps provide access to a broader prey range and higher hunting efficiency than smaller traps (see [27]). However, antlions are known for the establishment of dense 'antlion zones' [28,29], which suggests that at least under some conditions, certain factors might override the avoidance of high conspecific density.

Here, we performed a laboratory experiment that aimed to temporarily challenge larvae of *Euroleon nostras* antlions to either mild or harsh thermal stress and evaluated how the two types of experience affected subsequent trap construction under thermally non-challenging conditions. There is evidence suggesting that antlions originating from shaded areas in nature, such as our study species, might be more vulnerable to thermal challenges than antlions from open areas, such as *Myrmeleon bore* [30]. Nevertheless, antlions generally display extraordinary heat tolerance, and their foraging performance increases linearly with temperature [15,31–33]. Therefore, our experiment was preceded by measurements of the thermal performance of *E. nostras*, which helped us to empirically define mild and harsh thermal regimes for the studied animals. The effects of thermal experience have rarely been considered when studying trap construction in antlions. Here, we predicted that antlions that experienced harsh conditions would show decreased aggregation propensity—increasing neighbouring distance and building larger traps. In this way, antlions would compensate for the adverse physiological effects of the past. Ultimately, our work aimed to shed light on antlion decisions involved in trap construction, exploring whether thermal experience is one of the mediators in these decisions. Importantly, previous studies demonstrated similar mediation in other contexts in antlions, for example, in the case of starvation endurance and growth speed, both mediated by past prey availability [34,35].

## 2. Results

Over the course of one month under treatment conditions (mild, 25°C versus harsh, 31°C), during which time larvae were fed a single prey item once weekly, the majority of larvae exposed to 25°C maintained their initial body mass, whereas the majority of larvae exposed to 31°C lost some of their initial body mass (figure 1). In other words, individuals exposed to mild thermal treatment were in better condition than those exposed to harsh thermal treatment (table 1). When larvae were allowed to construct traps for 72 h after being transferred to common temperature conditions (20°C), they all relocated from their initial placement positions and built traps somewhat distant from each other. Larvae exposed to 31°C showed a nearly significant tendency ($F_{1,43} = 3.866$, $p = 0.054$) to build traps with a shorter neighbouring distance than larvae exposed to 25°C (figure 2). The analysis of trap diameters showed that the effect of the treatment alone was non-significant ($F_{1,85} = 1.718$, $p = 0.194$) and that larger larvae built larger traps ($F_{1,85} = 42.422$, $p < 0.001$). The two factors interacted

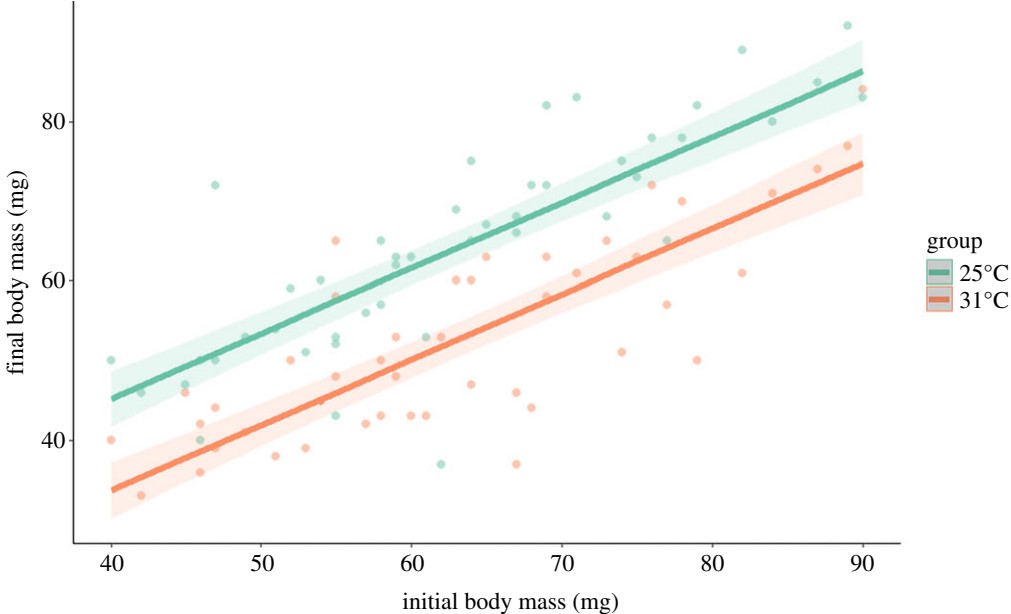

**Figure 1.** The final body mass of antlion larvae exposed to different treatments (mild, 25℃ versus harsh, 31℃) for one month in relation to their initial body mass. Larvae in both treatments were fed the same amount of prey. Green dots represent the mild treatment, and orange dots represent the harsh treatment. Lines represent model predictions, with shading indicating estimated confidence intervals.

**Table 1.** The results of the model considering the final body mass of antlion larvae exposed to different treatments (mild, 25℃ versus harsh, 31℃) for one month. The initial body mass represents the body mass before treatment, while the final body mass represents the body mass after treatment.

| model component | $F_{1,89}$ | deviance | residual deviance | $p$-value |
|---|---|---|---|---|
| null | | | 18498.0 | |
| treatment | 52.89 | 3053.26 | 15445.0 | <0.001 |
| initial body mass | 178.54 | 10307.0 | 5138.0 | <0.001 |

significantly ($F_{1,85} = 6.582$, $p = 0.012$), indicating that quantitative effects of larvae size depended on thermal treatment. In other words, smaller larvae built similarly sized traps regardless of their thermal exposure, but larger individuals that were exposed to 31℃ built smaller traps than larvae that were exposed to 25℃ (figure 3).

## 3. Discussion

Our data on body mass changes over the thermal treatment periods suggest that we succeeded in generating mild and harsh conditions for antlion larvae (figure 1). We found that at the end of the experiment, larvae originating from the harsh treatment had lower final body mass than those after the mild treatment, suggesting that the energy demand for withstanding the harsh condition was probably much higher than that required for withstanding the mild condition. This effect might indicate an increased channelling of resources to processes associated with coping with thermal stress, which was probably additionally enhanced by increased water evaporation in the harsh than mild conditions. Interestingly, when we compared absolute values of body masses before and after the experiment (see $X$ versus $Y$ values in figure 1), we found that not all animals gained in mass equally over the course of the experiment but that this depended on the initial body mass and the treatment. In particular, among the largest larvae, body mass losses occurred in both the mild and harsh treatment groups, although body mass loss was most severe in the latter group. By contrast, among the smallest larvae, those in the mild treatment group gained mass, while those in the harsh treatment

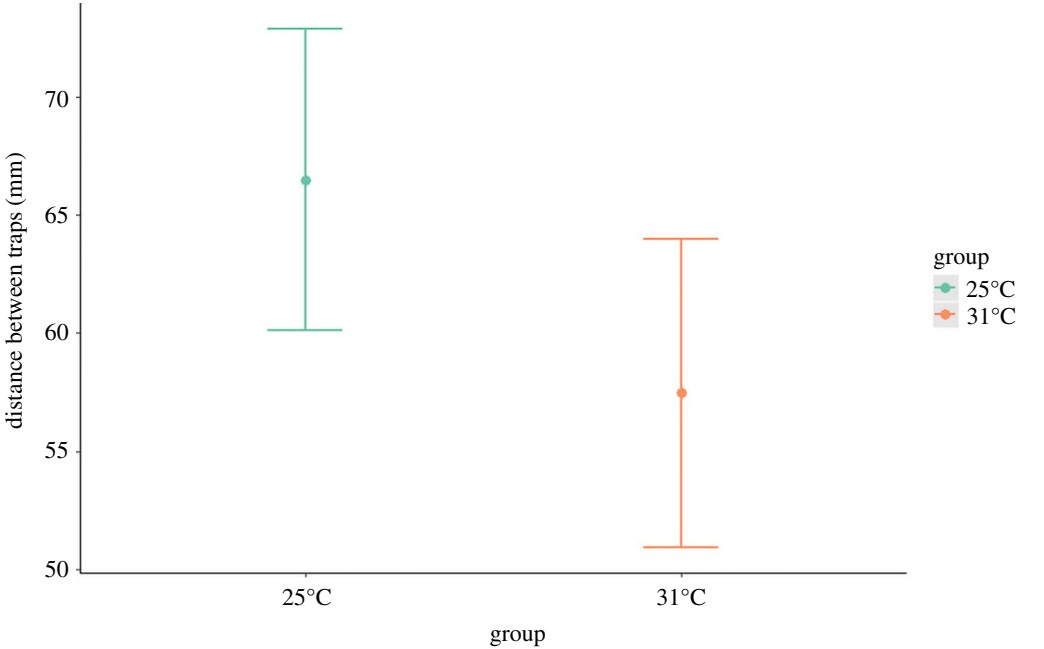

**Figure 2.** The distance between traps built under common temperature conditions (20°C) by antlion larvae previously exposed to different treatments (mild, 25°C versus harsh, 31°C). The figure shows the model predictions, with dots indicating means and whiskers indicating estimated confidence intervals.

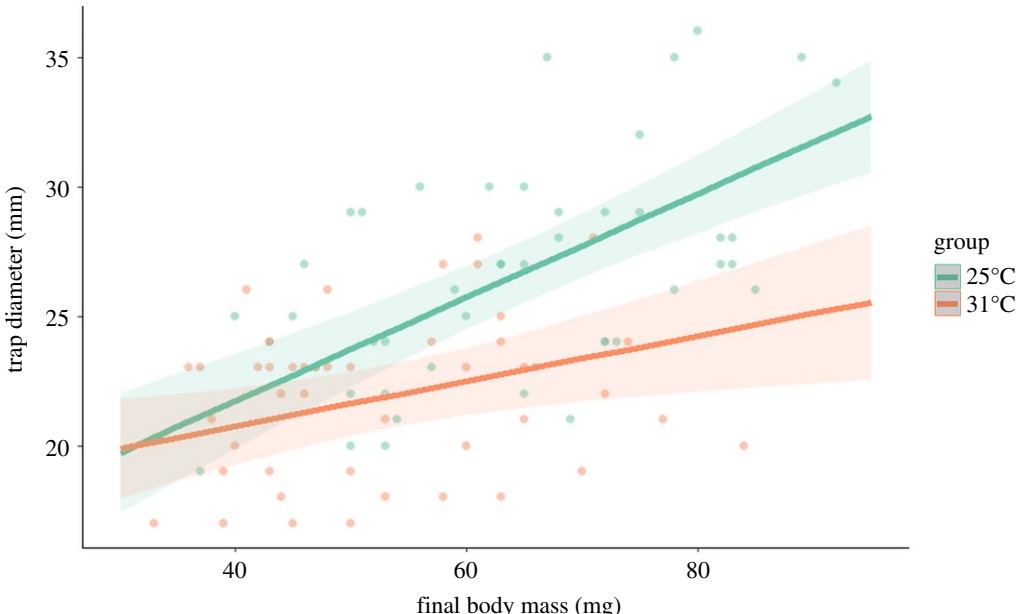

**Figure 3.** The diameter of traps built under common temperature conditions (20°C) by antlion larvae previously exposed to different treatments (mild, 25°C versus harsh, 31°C) in relation to their final body mass. Green dots represent the mild treatment, and orange dots represent the harsh treatment. Lines represent model predictions, with shading indicating estimated confidence intervals.

group suffered mass loss. This pattern suggests that the size of larvae in relation to the food regime at least partly determined how the larvae coped with the thermal treatments in our experiment. This effect certainly awaits further experimental investigation, perhaps with the involvement of a much wider range of body masses in larvae and different food regimes.

Different thermal exposures in larvae affected subsequent aggregation tendencies and trap sizes (figures 2 and 3). Specifically, larvae that were exposed to harsh thermal conditions settled *ca* 1 cm closer to neighbours (but note that the effect was only close to significance) than larvae that

experienced the mild treatment, and showed a weaker positive association between body mass and trap size. Both of these results were in contrast with our predictions about the induction of compensatory behaviours aimed at 'catching up' in growth and development after exposure to harsh conditions. Our predictions were tentatively based on the current knowledge about antlion aggregation behaviour; antlions residing close to each other often physically interfere with one another, leading to, for example, smaller trap sizes, even though larger traps are generally more efficient for prey capture [4,13,15,20,36,37]. That larger individuals, which were most severely affected by the harsh thermal experience, built smaller traps after exposure to 31°C than those exposed to 25°C, was possibly a natural consequence of the somewhat decreased neighbouring distance [15–17]. The pattern of trap building revealed in our study may indicate direct effects of exhaustion in larvae after harsh treatment [1]. Specifically, these individuals might have been less prone to investing energy into moving over longer distances (hence a trend towards decreased neighbouring distance) and trap construction (hence smaller traps), resembling, for example, antlions which start neglecting trap maintenance after a long period of starvation [37]. However, during exposure to an exhausting treatment, one can expect some increased mortality, which was not the case here (i.e. we observed no mortality at all in either group of individuals) and so makes this explanation uncertain. Alternatively, the detected changes may indicate attempts at compensation after experiencing harsh thermal conditions if we consider the so-called ricochet effect found in another group of trap builders, spiders [38,39]. Specifically, the capture of prey by a predator living in dense aggregation conditions with conspecifics may be facilitated by prey fatigue caused by unsuccessful hunting by conspecific predators. Aggregated antlions might also detect additional cues about incoming prey, e.g. sensing vibrations caused by hunting neighbours. Earlier research demonstrated that the response to prey depends on external factors in antlions [40] and that they might even, in the context of hunting, learn environmental signals (such as vibrations) and use them to their advantage [41–45]. Although hunting efficiency in antlions has been the subject of research interest, aggregation tendency as a way of changing hunting efficiency has never been studied. Both this and exhaustion, as potential drivers of behavioural changes such as those observed here, deserve further study.

In another group of trap builders, wormlions (Diptera: Vermileonidae), which represent an example of convergent evolution with antlions and sometimes even occur in sympatry [46,47], the effects of experience of various kinds are complex. For example, starvation periods sometimes increase the neighbouring distance in these dipterans but sometimes not, depending on other factors [48,49]. Abiotic conditions, such as previous medium quality or light exposure, most likely have little effect on the behaviours of wormlions [50–52]. This also applies to thermal experience, apparently, as the initial hindering effects of high temperature in terms of trap construction rates were diminishing over time in wormlions due to acclimatization [51]. The process of acclimatization nevertheless has its limits, as wormlion larvae kept at higher temperatures did not show an increased ability for heat shock recovery [53]. In our study, the detected differences in aggregation tendencies and trap sizes between treatment groups might be at least partly attributed to either more (25°C) or less (31°C) thermal acclimatization to our testing conditions (20°C). Studies in many ectotherms demonstrate rapid physiological acclimatization to new temperatures, even occurring within hours (e.g. 2 h in *Drosophila*, [54]). In our study, all the larvae were allowed to build traps for 72 h, which was probably long enough for all the antlions to acclimatize to the new thermal conditions. Consequently, we consider that the effects of differential acclimatization capacity were not responsible for the observed differences in trap building among our treatments. We stress that considering the long developmental time in antlions and wormlions, their experiences and acclimatization abilities need further research [30].

## 4. Conclusion

We showed here that previous thermal exposure influenced antlions' tendency to aggregate and the size of their traps to some extent. Thus, we achieved our ultimate goal and demonstrated that previously experienced thermal conditions might mediate trap construction behaviours in antlions. In species that prefer shaded and sheltered locations, including *E. nostras* used in the present study, the temperature effects on performance are well known [31,32,55], even if they are less dramatic than those in other species inhabiting open, sun-exposed locations (see [56,57]). It is very likely that in other more thermally tolerant species, such as the very thermophilic *Myrmeleon bore* [30,33], the effects of previous thermal exposure would be less pronounced. The extent to which our results are relevant to other sit-and-wait predators thus needs more comparative research attention.

# 5. Methods

In June 2020, we collected second instar larvae of *E. nostras* antlions in the Błędowska Desert (Poland, coordinates: 50°20′24′′ N, 19°32′20′′ E). This species is easily distinguished from other co-occurring species of antlions [58]. Following Devetak *et al.* [59], we differentiated instars on site based on body dimensions, mainly head size and shape. In the laboratory, we weighed the larvae to the nearest 0.001 g on an electronic balance (Mettler Toledo, Poland) to obtain an initial body mass. To limit the potential effects of body mass differences on aggregation tendency and trap size, we included only the most abundant size class of larvae, which ranged from 40 to 90 mg. Establishing two thermal conditions (mild versus harsh treatments), we reared 46 larvae at 25°C (mild treatment) and another 46 larvae at 31°C (harsh treatment) for one month. To avoid random bias in the average body mass of larvae under different treatments, we allocated larvae to treatments strictly controlling for body mass. For example, if a larva exposed to one treatment had a given body mass, a larva with a similar body mass was exposed to the other treatment. The thermal treatments were established in two thermal cabinets (Pol-Eco Aparatura, Poland) set to two constant temperatures. As larvae of *E. nostras* show shade preference and typically inhabit highly shaded microhabitats [7], the cabinets were set to constant darkness, which eliminated any potential effects of directional light on the larvae. Larvae were housed individually in labelled plastic cups (5 cm in diameter, 4 cm in height) half-filled with fine-grained (particle size of 250–500 μm) habitat-of-origin sand.

Two temperatures (25°C in the mild and 31°C in the harsh treatment) were chosen based on thermal performance assessments conducted in additionally collected second instar larvae of *E. nostras* of the same size class as indicated above. We conducted two tests ($N = 24$ in each); in the first, we recorded the temperatures preferred by larvae exposed to a thermal gradient [60], and in the second, we recorded the temperatures at which larvae displayed the loss of the righting reflex (LORR) when exposed to a steadily increasing temperature [61]. Both tests were conducted on the same day in a climatically controlled room (25°C, lit). In the first test, we exposed individual larvae to a thermal gradient inside an aluminium rail placed on a long platform (Biospekt, Poland), with an operational gradient inside the rail ranging from 22 to 44°C, with an approximately 2.0°C increase every 10 cm. We covered the bottom of the rail with a thin layer of dry sand, placed a single larva on the substrate in the rail at the hot end of the gradient and covered the rail with a transparent film for isolation. After 15 min, we evaluated the position of the larva along the gradient and recorded the temperature in the sand next to the animal to the nearest 0.1°C using a fast-response thermocouple thermometer (Delta OHM, Italy). In the second test, we exposed larvae to a steadily increasing temperature using a water bath (Memmert, Germany). We placed individual larvae into individual containers, which had a bottom covered with a thin layer of dry sand, and then placed the containers in a water bath (25°C). Then, we turned on the heat, which resulted in the temperature rising by approximately 1.2°C min$^{-1}$ until it reached 46°C. Using a thermocouple thermometer, we recorded the temperature in the sand occupied by the larva at the point when the larva showed LORR, which always followed bursts in mobility that accompanied an increase in temperature. The tests enabled us to calculate two mean temperatures, the first defining the preferred temperature in the gradient and the second defining the point of LORR in the water bath (mean ± s.d.: 35.6 ± 2.2°C and 41.1 ± 1.1°C, respectively). We used these statistics as the basis for establishing the temperatures in the thermal treatments, namely, 25°C for the mild treatment and 31°C for the harsh treatment.

Starting on the first day of the treatment experiment, we fed larvae once a week with a single *Lasius flavus* ant worker (length ranging 3–4 mm). All larvae actively hunted for the duration of the entire experiment, i.e. all had functional and undisturbed traps and responded to prey by catching it. After exposure to the treatments (no individual moulted or died during the study period), larvae were weighed to obtain their final body mass. Subsequently, we transferred all larvae to common conditions in the laboratory (20°C, natural photoperiod). The new conditions exposed all larvae to a lower temperature, which was either mildly lower (for those exposed to the mild treatment) or drastically lower (for those exposed to the harsh treatment). Upon transfer, the larvae originating from each treatment were randomly paired within a treatment group (23 pairs per treatment). Each pair was placed in an aluminium box (25 × 10 × 5 cm) half-filled with fine-grained sand (again, particle size of 250–500 μm and from the original habitat). The larvae were placed adjacent to each other in the middle of the box and were left undisturbed for 72 h, with the freedom to relocate and build traps. Then, we used an electronic calliper to measure (mm) the diameter of each trap and the distance from the middle of one trap to the middle of the other trap. Each larva was retrieved from the bottom of its trap and weighed again to identify individuals; individuals were identified based on their pre- and

post-placement body masses. One individual died during trap building (most likely harmed during the transfer), so the pair was excluded from further analyses. In total, we successfully tested 23 pairs exposed to mild treatment and 22 pairs exposed to harsh treatment.

Statistical analyses were conducted using the statistical programming language R [62] with the lmer4, ggplot2 and sjPlot packages. We first analysed the final body mass using a general linear model, with the treatment (25 versus 31°C) and the initial body mass as fixed factors. The model accounted for a treatment×initial body mass interaction, but it was found to be non-significant ($p = 0.244$) and removed. With another general linear model, we analysed the distance between traps, with the treatment (25 versus 31°C) and the absolute difference in body mass between larvae in a pair (body mass difference) as fixed factors. The model also accounted for a treatment×body mass difference interaction. However, the body mass difference and interaction were non-significant ($p = 0.589$ and $p = 0.137$, respectively); hence, both were removed. Finally, we used a generalized linear mixed model to analyse the trap diameters, with the treatment (25 versus 31°C) and the final body mass as fixed factors and the box with a pair of larvae as the random factor. The model accounted for a treatment×final body mass interaction.

Data accessibility. The data are provided in electronic supplementary material [63].

Authors' contributions. K.M. conceived of the study, designed the study, conducted the experiments, carried out the statistical analyses, interpreted the results and drafted the manuscript. M.C. carried out the statistical analyses, interpreted the results and drafted the manuscript. Both authors gave final approval for publication and agree to be held accountable for the work performed therein.

Competing interests. We declare we have no competing interests.

Funding. This work was supported by the National Science Centre in Poland (grant SONATINA 3, number 2019/32/C/NZ8/00128).

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
