## [Peer Review File · Royal Society Open Science]

Review History

RSOS-210163.R0 (Original submission)

Review form: Reviewer 1

Is the manuscript scientifically sound in its present form?

Yes

Are the interpretations and conclusions justified by the results?

Yes

Is the language acceptable?

Yes

Do you have any ethical concerns with this paper?

No

Have you any concerns about statistical analyses in this paper?

No

Recommendation?

Major revision is needed (please make suggestions in comments)

Comments to the Author(s)

I have read the manuscript entitled "Past thermal conditions affect hunting behaviour in larval antlions" submitted to the Royal Society Open Science. The authors kept antlion larvae under two temperatures, higher/harsh and lower/benign for a month. They first demonstrate that antlions in the harsh temperature lost more mass than those in the benign temperature. Then, they placed two antlions together, one adjacent to another, under benign temperature and let them construct pit-traps. Those previously exposed to benign temperature constructed larger pit-traps and tended to move longer distances away from the other individual. I find the results interesting and agree with the authors that no one before examined the effects of previous thermal conditions on antlion behavior when returned to benign conditions. In general, examining the carry-over effects of previous maintenance conditions is a timely question and we do not know enough on such consequences. The experiment is generally well conducted, the paper nicely and clearly written. It is a small study, and the paper is therefore concisely written, which should be much appreciated. I have several main comments, which I will summarize here. For more specific comments, please see below. (1) The distance between individuals is not significant but is only a trend. Yet, it is described as a strong, significant result in the Discussion and Abstract. Please be more careful. (2) We do not know whether the distance between individuals is a result of competition, or any other interaction, or whether it is a consequence of exhaustion, or weak individual physiological state caused by the previous thermal treatment. What is missing here is a control with the distance covered by individual larvae exposed to benign temperature after kept in both benign and harsh temperatures. Alternatively, could you provide data for individual movements of the two antlions from the initial placement location? I would expect then that large individuals would move more than small ones. Individual movement distances of the two antlions are more informative than the final distance between the two individuals, which is harder to interpret. (3) I think that the consequences of exposure to stress can go in both ways. While mild stress sometimes improve performance and leads to acclimation, harsh stress often has the opposite effect. This should be stressed out in the Introduction and the predictions could be also rephrased. Search for the term "hormesis" and perhaps refer to it.

Lines 44-46: I think that most evidence points to elevated relocation when density increases. See for example, Griffiths (1992 *Ecol Entomol*), Day and Zalucki (2000 *Austral Ecology*).

Lines 48-51: Maybe a more accurate way to phrase it would be to write that while antlions prefer to avoid high conspecific density when possible, abiotic requirements, such as available substrate, often override the negative effects of density.

Lines 57-60: Harsh temperatures might also have the opposite effect, as they induce stress, make the antlions weaker and prevent them from building large pits (to save the already limited energy).

Line 66: What do you mean by a "confounding factor"? Unclear.

Introduction in general: As I see it, the study focuses on three main topics: (1) the effect of temperature on pit construction; (2) the effect of past conditions on current performance; and (3) tendency to aggregate or space out. The third topic is well covered, but in my opinion, you can elaborate more on the first two topics. For example, how does temperature affect pit construction in pit-building predators? (here I would refer also to wormlions). How does temperature affect relocation decisions? How do past conditions affect current behavior in pit-builders?

Lines 76-78 and the associated statistics: The effect of the mass difference as well as mass difference x treatment are not significant. Please remove them from the test and redo the test. Then, you can get a better idea if temperature really has an effect or not. Also, in Table 2, I think the df is wrong – the error df should be 41 instead of 44, as your sample size is 45, and you "lose" one df for each of those: the thermal treatment, the mass difference, and the interaction. Please recheck the dfs in your statistics, or maybe I missed here something. Finally, I would add the effect size here, to convince the readers the results are meaningful, even if only marginally significant.

Lines 82-85: This result fits my expectation that high temperature would induce some stress leading to smaller pits constructed.

Line 98: Please mention earlier that the larvae were fed and how frequently.

Lines 102-105: Please be more careful here. You showed a tendency to space-out when previously kept at high temperature. The difference was not significant. Please take care that a trend in the Results will not become a solid evidence in the Discussion. The same apparently strong result is described in the Abstract, which does not match the results in 100%.

Lines 105-107: The effect of stress can go in both ways. The response to mild, short-term stress could be "beneficial". Have a look for the term "hormesis" and papers by Le Bourg. The response to harsher and longer stress are usually negative, making the animals weaker.

Lines 113-116: This sounds a bit far-fetched to me. No one demonstrated the "ricochet effect" in pit-builders. I think it is almost always better to be far away from competitors and construct large pit. Furthermore, you ignore here the heavy cost of cannibalism, which is common in antlions, and is another argument why not residing too close to each other.

Discussion: What is in your opinion the reason for spacing out in response to high temperature? I would discuss the mechanism.

Line 170-171: So perform a t-test on the initial body mass of antlions allocated to the two treatments and show it has a very high p value.

Lines 174-175: Keeping the antlions under constant dark conditions may lead to the construction of smaller pits, as shown in Scharf et al. (2008 Animal Behaviour). I think that a natural light:dark cycle is better and presents better the natural conditions of antlions.

Lines 176-177: So the sand depth was 2 cm? Could it be that while small individuals are not limited by this sand depth, large individuals are? If only large individuals are limited by sand depth, can it affect your results in some way? Also, what do you mean by fin-grained sand? Can you be more accurate and give a range of particle sizes? Was the sand collected from the natural habitat?

Line 205: What is the typical/average size of the ant?

Lines 213-214: Similar question here – could it be that 2.5 cm sand depth limits the ability of large individuals to construct sufficiently large pits?

Line 214: next to each other – adjacent to each other?

Lines 214-217: As you do not have a control of the distance moved by single individuals, you cannot determine whether the distances moved by the two individuals are due to density-related reasons (i.e., trying to get away from each other), or individual reasons (i.e., higher metabolic rate). It seems like a control group of single individuals would have been helpful here.

Lines 222-231: I would remove non-significant interaction terms and redo the test. It might be that the main effects get more significant (or sometimes less significant) after such removal.

Review form: Reviewer 2

Is the manuscript scientifically sound in its present form?

Yes

Are the interpretations and conclusions justified by the results?

Yes

Is the language acceptable?

Yes

Do you have any ethical concerns with this paper?

No

Have you any concerns about statistical analyses in this paper?

No

Recommendation?

Major revision is needed (please make suggestions in comments)

Comments to the Author(s)

Please see the attached file with my comments (Appendix A). Thank you

Decision letter (RSOS-210163.R0)

Dear Mr Miler

The Editors assigned to your paper RSOS-210163 "Past thermal conditions affect hunting behaviour in larval antlions" have now received comments from reviewers and would like you to revise the paper in accordance with the reviewer comments and any comments from the Editors. Please note this decision does not guarantee eventual acceptance.

Please submit your revised manuscript and required files (see below) no later than 21 days from today's (ie 04-May-2021) date. Note: the ScholarOne system will 'lock' if submission of the revision is attempted 21 or more days after the deadline. If you do not think you will be able to meet this deadline please contact the editorial office immediately.

on behalf of Dr Kimberley Mathot (Associate Editor) and Kevin Padian (Subject Editor)
openscience@royalsociety.org

Associate Editor Comments to Author (Dr Kimberley Mathot):

Comments to the Author:

Dear Mr. Miller,

I apologize for the delay in reaching a decision on your submission "Past thermal conditions affect hunting behaviour in larval antlions". It took quite some time to find available and qualified referees. I am happy to be able to report that we have now received reviews from two experts in the field, who are both very positive about the work. The both raise a number of points that they would like to see addressed in revisions, and I am therefore recommending Major Revisions.

Specifically, please be ensure you are consistent about how you interpret results that are not statistically significant if you are setting the criteria for significance as $p < 0.05$. Personally, I am not a fan of arbitrary p-values, so I have no problem with discussing results that don't reach this level of significance. But then the rationale for referring to something as biologically important or not needs to be explicit (e.g., based on the estimated effect size). Referee #1 recommends using model reduction to clarify the importance of some of the main effects- but I would specifically ask that you not do this. While it will likely make marginally significant p-values significant by reducing the number of parameters estimated, it won't change the biological interpretation. Thus, for transparency, and to avoid p-hacking, I would recommend you maintain the model structures as they are and instead focus on effect sizes in your interpretation of the results.

Referee #1 also makes important points about uncertainty regarding the mechanism underlying the distance effect (general point 2), and potential non-linear effects of stress (general point 3), and referee #2 asks for more careful attention to the statements regarding the generality of the results. Please consider addressing these in the revised version, as I feel they are important points.

Finally, both referees have provided a list of minor corrections that could be implemented.

Thank you for submitting your work for consideration at RSOS. We look forward to receiving your revised manuscript.

Reviewer comments to Author:

Reviewer: 1

Comments to the Author(s)

I have read the manuscript entitled "Past thermal conditions affect hunting behaviour in larval antlions" submitted to the Royal Society Open Science. The authors kept antlion larvae under two temperatures, higher/harsh and lower/benign for a month. They first demonstrate that antlions in the harsh temperature lost more mass than those in the benign temperature. Then, they placed two antlions together, one adjacent to another, under benign temperature and let them construct pit-traps. Those previously exposed to benign temperature constructed larger pit-traps and tended to move longer distances away from the other individual. I find the results interesting and agree with the authors that no one before examined the effects of previous thermal conditions on antlion behavior when returned to benign conditions. In general, examining the carry-over effects of previous maintenance conditions is a timely question and we do not know enough on such consequences. The experiment is generally well conducted, the paper nicely and clearly written.

It is a small study, and the paper is therefore concisely written, which should be much appreciated. I have several main comments, which I will summarize here. For more specific comments, please see below. (1) The distance between individuals is not significant but is only a trend. Yet, it is described as a strong, significant result in the Discussion and Abstract. Please be more careful. (2) We do not know whether the distance between individuals is a result of competition, or any other interaction, or whether it is a consequence of exhaustion, or weak individual physiological state caused by the previous thermal treatment. What is missing here is a control with the distance covered by individual larvae exposed to benign temperature after kept in both benign and harsh temperatures. Alternatively, could you provide data for individual movements of the two antlions from the initial placement location? I would expect then that large individuals would move more than small ones. Individual movement distances of the two antlions are more informative than the final distance between the two individuals, which is harder to interpret. (3) I think that the consequences of exposure to stress can go in both ways. While mild stress sometimes improve performance and leads to acclimation, harsh stress often has the opposite effect. This should be stressed out in the Introduction and the predictions could be also rephrased. Search for the term "hormesis" and perhaps refer to it.

Lines 44-46: I think that most evidence points to elevated relocation when density increases. See for example, Griffiths (1992 *Ecol Entomol*), Day and Zalucki (2000 *Austral Ecology*).

Lines 48-51: Maybe a more accurate way to phrase it would be to write that while antlions prefer to avoid high conspecific density when possible, abiotic requirements, such as available substrate, often override the negative effects of density.

Lines 57-60: Harsh temperatures might also have the opposite effect, as they induce stress, make the antlions weaker and prevent them from building large pits (to save the already limited energy).

Line 66: What do you mean by a "confounding factor"? Unclear.

Introduction in general: As I see it, the study focuses on three main topics: (1) the effect of temperature on pit construction; (2) the effect of past conditions on current performance; and (3) tendency to aggregate or space out. The third topic is well covered, but in my opinion, you can elaborate more on the first two topics. For example, how does temperature affect pit construction in pit-building predators? (here I would refer also to wormlions). How does temperature affect relocation decisions? How do past conditions affect current behavior in pit-builders?

Lines 76-78 and the associated statistics: The effect of the mass difference as well as mass difference x treatment are not significant. Please remove them from the test and redo the test. Then, you can get a better idea if temperature really has an effect or not. Also, in Table 2, I think the df is wrong - the error df should be 41 instead of 44, as your sample size is 45, and you "lose" one df for each of those: the thermal treatment, the mass difference, and the interaction. Please recheck the dfs in your statistics, or maybe I missed here something. Finally, I would add the effect size here, to convince the readers the results are meaningful, even if only marginally significant.

Lines 82-85: This result fits my expectation that high temperature would induce some stress leading to smaller pits constructed.

Line 98: Please mention earlier that the larvae were fed and how frequently.

Lines 102-105: Please be more careful here. You showed a tendency to space-out when previously kept at high temperature. The difference was not significant. Please take care that a trend in the Results will not become a solid evidence in the Discussion. The same apparently strong result is described in the Abstract, which does not match the results in 100%.

Lines 105-107: The effect of stress can go in both ways. The response to mild, short-term stress could be "beneficial". Have a look for the term "hormesis" and papers by Le Bourg. The response to harsher and longer stress are usually negative, making the animals weaker.

Lines 113-116: This sounds a bit far-fetched to me. No one demonstrated the "ricochet effect" in pit-builders. I think it is almost always better to be far away from competitors and construct large

pit. Furthermore, you ignore here the heavy cost of cannibalism, which is common in antlions, and is another argument why not residing too close to each other.

Discussion: What is in your opinion the reason for spacing out in response to high temperature? I would discuss the mechanism.

Line 170-171: So perform a t-test on the initial body mass of antlions allocated to the two treatments and show it has a very high p value.

Lines 174-175: Keeping the antlions under constant dark conditions may lead to the construction of smaller pits, as shown in Scharf et al. (2008 Animal Behaviour). I think that a natural light:dark cycle is better and presents better the natural conditions of antlions.

Lines 176-177: So the sand depth was 2 cm? Could it be that while small individuals are not limited by this sand depth, large individuals are? If only large individuals are limited by sand depth, can it affect your results in some way? Also, what do you mean by fin-grained sand? Can you be more accurate and give a range of particle sizes? Was the sand collected from the natural habitat?

Line 205: What is the typical/average size of the ant?

Lines 213-214: Similar question here – could it be that 2.5 cm sand depth limits the ability of large individuals to construct sufficiently large pits?

Line 214: next to each other – adjacent to each other?

Lines 214-217: As you do not have a control of the distance moved by single individuals, you cannot determine whether the distances moved by the two individuals are due to density-related reasons (i.e., trying to get away from each other), or individual reasons (i.e., higher metabolic rate). It seems like a control group of single individuals would have been helpful here.

Lines 222-231: I would remove non-significant interaction terms and redo the test. It might be that the main effects get more significant (or sometimes less significant) after such removal.

Reviewer: 2

Comments to the Author(s)

Please see the attached file with my comments. Thank you

===PREPARING YOUR MANUSCRIPT===

===PREPARING YOUR REVISION IN SCHOLARONE===

Author's Response to Decision Letter for (RSOS-210163.R0)

See Appendix B.

Decision letter (RSOS-210163.R1)

Dear Mr Miler,

It is a pleasure to accept your manuscript entitled "Past thermal conditions affect hunting behaviour in larval antlions" in its current form for publication in Royal Society Open Science.

on behalf of Dr Kimberley Mathot (Associate Editor) and Kevin Padian (Subject Editor)
openscience@royalsociety.org

Associate Editor Comments to Author (Dr Kimberley Mathot):
Associate Editor
Comments to the Author:
Dear Dr. Miler,

In reviewing your revised MS and associated files, I realize that in my last correspondence with you I had a typo in the spelling of your name, and also used your incorrect title. I apologize for that.

I have now reviewed your response to referees and the associated changes in the manuscript and am satisfied that you have addressed the comments raised in the initial review. I am happy to recommend your manuscript be accepted for publication.

Thank you for submitting your work to Royal Society Open Science.

Appendix A

The authors performed lab experiments that tested whether different thermal conditions would affect aggregation behavior and trap size in antlion larvae. The manuscript is written well and the results are interesting.

My major concern is that these experiments indicate on a relatively secluded case. In literature predictions were shown to reaffirm, while here – on the contrary. I believe it is still publishable but perhaps with less tendency to extend the conclusions more broadly. Please rephrase.

Minor comments:

Results

Lines 70-72: Loosing body weight in the harsh condition while not loosing it in the mild condition may suggest that the harsh condition is not within the optimal temperature limit of the antlion larvae.

Line 77: Almost significant difference between the two conditions (harsh and mild) in distance between neighbors for building traps. Harsh – less distance. In contrary to predictions, although only almost significant.

Lines 83-84: smaller larvae built similarly sized traps regardless of their thermal exposure, but larger individuals that were exposed to 31°C built smaller traps than larvae that were exposed to 25°C (significant)- your most interesting and important result. I would add more information and elaborate more on past studies' results and compare with yours (in discussion).

Line 85: the effect of the treatment alone was nonsignificant. Please explain why you think you got this result in the discussion.

Discussion

Lines 94-95 – Perhaps larvae on both conditions lost weight due to some rearing effect in lab? Regardless of which thermal condition they were assigned to?

Line 153-157: please rephrase. Your treatment effect was non-significant. You can relate to the significant effect of body size x treatment.

Methods

From line 178: I suggest giving this experiment attention. You can reframe the article with giving this one the name experiment 1 and experiment 2, to the thermal conditions experiment. In my eyes, experiment 1 is equally important. You do not mention this experiment in the Results section.

I thought initially that the reason you might have not gotten a significant result for treatment in experiment 2, is because the difference between the conditions was not strong enough for the larvae. The first experiment show you have carefully tested which conditions are harsh or mild for the larvae. I would emphasize this.

Figures:

In all, I was unable to read the headlines of the y axis ! Please correct

Appendix B

Dear Mr. Miller,

I apologize for the delay in reaching a decision on your submission "Past thermal conditions affect hunting behaviour in larval antlions". It took quite some time to find available and qualified referees. I am happy to be able to report that we have now received reviews from two experts in the field, who are both very positive about the work. The both raise a number of points that they would like to see addressed in revisions, and I am therefore recommending Major Revisions.

Specifically, please be ensure you are consistent about how you interpret results that are not statistically significant if you are setting the criteria for significance as p
Referee #1 also makes important points about uncertainty regarding the mechanism underlying the distance effect (general point 2), and potential non-linear effects of stress (general point 3), and referee #2 asks for more careful attention to the statements regarding the generality of the results. Please consider addressing these in the revised version, as I feel they are important points.

Finally, both referees have provided a list of minor corrections that could be implemented.

Thank you for submitting your work for consideration at RSOS. We look forward to receiving your revised manuscript.

>>> Thank you for giving us the opportunity to revise our work. We provide responses to each point raised in the reviews – please see below. Issues you point out are all addressed. Specifically, we softened our claims regarding the generality and importance of our nearly significant result, and we expressed uncertainty regarding the reasons behind it.

Reviewer comments to Author:

Reviewer: 1

I have read the manuscript entitled "Past thermal conditions affect hunting behaviour in larval antlions" submitted to the Royal Society Open Science. The authors kept antlion larvae under two temperatures, higher/harsh and lower/benign for a month. They first demonstrate that antlions in the harsh temperature lost more mass than those in the benign temperature. Then, they placed two antlions together, one adjacent to another, under benign temperature and let them construct pit-traps. Those previously exposed to benign temperature constructed larger pit-traps and tended to move longer distances away from the other individual. I find the results interesting and agree with the authors that no one before examined the effects of previous thermal conditions on antlion behavior when returned to benign conditions. In general, examining the carry-over effects of previous maintenance conditions is a timely question and we do not know enough on such consequences. The experiment is generally well conducted, the paper nicely and clearly written. It is a small study, and the paper is therefore concisely written, which should be much appreciated. I have several main comments, which I will summarize here. For more specific comments, please see below.

>>> Thank you for your effort to improve our manuscript.

(1) The distance between individuals is not significant but is only a trend. Yet, it is described as a strong, significant result in the Discussion and Abstract. Please be more careful.

>>> We softened our claim throughout the manuscript.

(2) We do not know whether the distance between individuals is a result of competition, or any other interaction, or whether it is a consequence of exhaustion, or weak individual physiological state caused by the previous thermal treatment. What is missing here is a control with the distance covered by individual larvae exposed to benign temperature after kept in both benign and harsh temperatures. Alternatively, could you provide data for individual movements of the two antlions from the initial placement location? I would expect then that large individuals would move more than small ones. Individual movement distances of the two antlions are more informative than the final distance between the two individuals, which is harder to interpret.

>>> We now note explicitly that the reason for the trend is uncertain (see lines 109-140).

(3) I think that the consequences of exposure to stress can go in both ways. While mild stress sometimes improve performance and leads to acclimation, harsh stress often has the opposite effect. This should be stressed out in the Introduction and the predictions could be also rephrased. Search for the term "hormesis" and perhaps refer to it.

>>> We expanded our discussion of the results and indicate that they might stem from high stress of individuals after the harsh treatment (see lines 109-140).

Lines 44-46: I think that most evidence points to elevated relocation when density increases. See for example, Griffiths (1992 Ecol Entomol), Day and Zalucki (2000 Austral Ecology).

>>> Yes, we indicate that most evidence points to high density as being unfavorable for antlions (see lines 38-43).

Lines 48-51: Maybe a more accurate way to phrase it would be to write that while antlions prefer to avoid high conspecific density when possible, abiotic requirements, such as available substrate, often override the negative effects of density.

>>> Agreed and rephrased (see lines 43-52).

Lines 57-60: Harsh temperatures might also have the opposite effect, as they induce stress, make the antlions weaker and prevent them from building large pits (to save the already limited energy).

>>> This we now explicitly write in the discussion (see lines 109-140).

Line 66: What do you mean by a "confounding factor"? Unclear.

>>> Rephrased (see lines 43-52, 66-70).

Introduction in general: As I see it, the study focuses on three main topics: (1) the effect of temperature on pit construction; (2) the effect of past conditions on current performance; and (3) tendency to aggregate or space out. The third topic is well covered, but in my opinion, you can elaborate more on the first two topics. For example, how does temperature affect pit construction in pit-building predators? (here I would refer also to wormlions). How does temperature affect relocation decisions? How do past conditions affect current behavior in pit-builders?

>>> Agreed, we added information about the effects of temperature (see lines 53-63) and it is also mentioned in conclusions (see lines 163-172). We omit the issue of the effect of past conditions on current performance, except for a brief mention (see lines 66-70) as you noted yourself that it is best to keep the manuscript concise and we elaborate on that subject more in the discussion (see lines 109-160).

Lines 76-78 and the associated statistics: The effect of the mass difference as well as mass difference \times treatment are not significant. Please remove them from the test and redo the test. Then, you can get a better idea if temperature really has an effect or not.

>>> We removed nonsignificant terms from models (see lines 239-251) and re-run them to report the results and produce the figures (see lines 73-88 and Tab. 1).

Also, in Table 2, I think the df is wrong – the error df should be 41 instead of 44, as your sample size is 45, and you "lose" one df for each of those: the thermal treatment, the mass difference, and the interaction. Please recheck the dfs in your statistics, or maybe I missed here something. Finally, I would add the effect size here, to convince the readers the results are meaningful, even if only marginally significant.

>>> Thank you for noticing this issue, we have somehow reported incorrect dfs. This is now corrected throughout. We also noted the difference between the two groups, which can be visible in Fig. 2 (see lines 80-82, 109-113 and Fig. 2).

Lines 82-85: This result fits my expectation that high temperature would induce some stress leading to smaller pits constructed.

>>> We now refer to this potential explanation in the discussion (see lines 109-140).

Line 98: Please mention earlier that the larvae were fed and how frequently.

>>> Agreed and done (see lines 73-76).

Lines 102-105: Please be more careful here. You showed a tendency to space-out when previously kept at high temperature. The difference was not significant. Please take care that a trend in the Results will not become a solid evidence in the Discussion. The same apparently strong result is described in the Abstract, which does not match the results in 100%.

>>> Agreed and rephrased throughout the manuscript.

Lines 105-107: The effect of stress can go in both ways. The response to mild, short-term stress could be "beneficial". Have a look for the term "hormesis" and papers by Le Bourg. The response to harsher and longer stress are usually negative, making the animals weaker.

>>> Hormesis relates primarily to health and longevity, and we feel this is not truly relevant here. We refer to possible higher stress levels in larvae after the harsh treatment as driving the following differences (see lines 109-140).

Lines 113-116: This sounds a bit far-fetched to me. No one demonstrated the "ricochet effect" in pit-builders. I think it is almost always better to be far away from competitors and construct large pit. Furthermore, you ignore here the heavy cost of cannibalism, which is common in antlions, and is another argument why not residing too close to each other.

>>> We leave this as a suggestion, a possible alternative explanation (see lines 109-140).

Discussion: What is in your opinion the reason for spacing out in response to high temperature? I would discuss the mechanism.

>>> We added a possible reason, related to earlier stress and later exhaustion (see lines 109-140).

Line 170-171: So perform a t-test on the initial body mass of antlions allocated to the two treatments and show it has a very high p value.

>>> There is no need as these two groups were weight-matched and practically identical regarding their initial body mass – that was the aim of the procedure.

Lines 174-175: Keeping the antlions under constant dark conditions may lead to the construction of smaller pits, as shown in Scharf et al. (2008 Animal Behaviour). I think that a natural light:dark cycle is better and presents better the natural conditions of antlions.

Lines 176-177: So the sand depth was 2 cm? Could it be that while small individuals are not limited by this sand depth, large individuals are? If only large individuals are limited by sand depth, can it affect your results in some way?

>>> Yes, larvae were kept under constant dark and with ~2 cm of sand, but this was during the "rearing stage", when all larvae (from both groups) were kept in the cups in the thermal

cabinets. Measurements (e.g. of trap size) were performed later, after all larvae were transferred out of these cabinets. See below for our response about the sand layer thickness.

Also, what do you mean by fin-grained sand? Can you be more accurate and give a range of particle sizes? Was the sand collected from the natural habitat?

>>> Yes, this was the habitat-of-origin sand. We added the information (see lines 191-193).

Line 205: What is the typical/average size of the ant?

>>> Workers are about 3-4 mm in length. We added the information (see lines 220-221).

Lines 213-214: Similar question here – could it be that 2.5 cm sand depth limits the ability of large individuals to construct sufficiently large pits?

>>> It probably would in third instars, but we studied second instars and majority of them rather small in size. We don't believe this was a problem, e.g. we never saw the bottom of the trap to be at the bottom of the box.

Line 214: next to each other – adjacent to each other?

>>> Agreed and changed (see lines 231-232).

Lines 214-217: As you do not have a control of the distance moved by single individuals, you cannot determine whether the distances moved by the two individuals are due to density-related reasons (i.e., trying to get away from each other), or individual reasons (i.e., higher metabolic rate). It seems like a control group of single individuals would have been helpful here.

>>> We noted this shortcoming of the study in the discussion (see lines 109-140).

Lines 222-231: I would remove non-significant interaction terms and redo the test. It might be that the main effects get more significant (or sometimes less significant) after such removal.

>>> Done (see lines 239-251). Removal changed the results only slightly.

Reviewer: 2

Comments to the Author(s)

The authors performed lab experiments that tested whether different thermal conditions would affect aggregation behavior and trap size in antlion larvae. The manuscript is written well and the results are interesting. My major concern is that these experiments indicate on a relatively secluded case. In literature predictions were shown to reaffirm, while here – on the contrary. I believe it is still publishable but perhaps with less tendency to extend the conclusions more broadly. Please rephrase.

>>> Thank you for your comments. We modified the text to be more cautious (see lines 15-16, 171-172).

Minor comments:

Results

Lines 70-72: Loosing body weight in the harsh condition while not loosing it in the mild condition may suggest that the harsh condition is not within the optimal temperature limit of the antlion larvae.

>>> Indeed, this was the aim here, to stress individuals in the harsh condition, but not in the mild condition (see lines 53-70, 194-219).

Line 77: Almost significant difference between the two conditions (harsh and mild) in distance between neighbors for building traps. Harsh – less distance. In contrary to predictions, although only almost significant.

>>> We modified the text to be more careful and make it clear that this result is only a trend (see lines 11-13, 78-82, 163-164).

Lines 83-84: smaller larvae built similarly sized traps regardless of their thermal exposure, but larger individuals that were exposed to 31°C built smaller traps than larvae that were exposed to 25°C (significant) - your most interesting and important result. I would add more information and elaborate more on past studies' results and compare with yours (in discussion).

>>> This is difficult as previous studies scarcely focused on how antlions react to previous thermal conditions.

Line 85: the effect of the treatment alone was nonsignificant. Please explain why you think you got this result in the discussion.

>>> Perhaps because majority of our larvae was rather small and so built relatively small traps. This can be seen in Fig. 3. We note in the discussion that it would be beneficial to study a broader range of larvae sizes (106-108).

Discussion

Lines 94-95 – Perhaps larvae on both conditions lost weight due to some rearing effect in lab? Regardless of which thermal condition they were assigned to?

>>> This might have been related to the food regime, sufficient for smaller but not larger individuals (see lines 98-108).

Line 153-157: please rephrase. Your treatment effect was non-significant. You can relate to the significant effect of body size x treatment.

>>> Rephrased (see lines 163-172).

Methods

From line 178: I suggest giving this experiment attention. You can reframe the article with giving this one the name experiment 1 and experiment 2, to the thermal conditions experiment. In my eyes, experiment 1 is equally important. You do not mention this experiment in the Results section.

>>> We decided to keep this part as an introductory element of the methods, helping us to establish the thermal regimes, but we note in the beginning of the text that we performed such testing for more accuracy (see lines 53-70).

I thought initially that the reason you might have not gotten a significant result for treatment in experiment 2, is because the difference between the conditions was not strong enough for the larvae. The first experiment show you have carefully tested which conditions are harsh or mild for the larvae. I would emphasize this.

>>> Yes, we tried to fit these conditions to our study species, and larvae were exposed to those for a month. It's difficult to tell at this point why the results were not more pronounced.

Figures:

In all, I was unable to read the headlines of the y axis! Please correct.

>>> We provided new figures with increased readability (please see Fig. 1, 2 & 3).